# Bacterial Communities of Forest Soils along Different Elevations: Diversity, Structure, and Functional Composition with Potential Impacts on CO_2_ Emission

**DOI:** 10.3390/microorganisms10040766

**Published:** 2022-04-01

**Authors:** Wanlong Sun, Zhouyuan Li, Jiesi Lei, Xuehua Liu

**Affiliations:** 1State Key Joint Laboratory of Environmental Simulation and Pollution Control, School of Environment, Tsinghua University, Beijing 100084, China; wlsun@tsinghua.edu.cn (W.S.); leijs20@mails.tsinghua.edu.cn (J.L.); 2China Grassland Research Center, School of Grassland Science, Beijing Forestry University, Beijing 100083, China; lizhouyuan@bjfu.edu.cn

**Keywords:** bacterial communities, functional annotation, CO_2_ effluxes, elevation, 16S rRNA

## Abstract

Soil bacteria are important components of forest ecosystems, there compostion structure and functions are sensitive to environmental conditions along elevation gradients. Using 16S rRNA gene amplicon sequencing followed by FAPROTAX function prediction, we examined the diversity, composition, and functional potentials of soil bacterial communities at three sites at elevations of 1400 m, 1600 m, and 2200 m in a temperate forest. We showed that microbial taxonomic composition did not change with elevation (*p* = 0.311), though soil bacterial α-diversities did. *Proteobacteria*, *Acidobacteria*, *Actinobacteria*, and *Verrucomicrobia* were abundant phyla in almost all soil samples, while *Nitrospirae*, closely associated with soil nitrogen cycling, was the fourth most abundant phylum in soils at 2200 m. Chemoheterotrophy and aerobic chemoheterotrophy were the two most abundant functions performed in soils at 1400 m and 1600 m, while nitrification (25.59% on average) and aerobic nitrite oxidation (19.38% on average) were higher in soils at 2200 m. Soil CO_2_ effluxes decreased (*p* < 0.050) with increasing elevation, while they were positively correlated (r = 0.55, *p* = 0.035) with the abundances of bacterial functional groups associated with carbon degradation. Moreover, bacterial functional composition, rather than taxonomic composition, was significantly associated with soil CO_2_ effluxes, suggesting a decoupling of taxonomy and function, with the latter being a better predictor of ecosystem functions. Annual temperature, annual precipitation, and pH shaped (*p* < 0.050) both bacterial taxonomic and functional communities. By establishing linkages between bacterial taxonomic communities, abundances of bacterial functional groups, and soil CO_2_ fluxes, we provide novel insights into how soil bacterial communities could serve as potential proxies of ecosystem functions.

## 1. Introduction

Soil microorganisms are important members of forest ecosystems [1]. Soil microbes (especially nitrogen-fixing bacteria and fungi belonging to the phyla *Glomeromycota*, *Basidiomycota*, and *Ascomycota*) enable plants to accumulate nitrogen, phosphorus, and sulfur, which are essential for forest-living organisms in the soil and accelerate nutrient cycling through organic matter decomposition and nitrogen-fixing [2,3]. On the one hand, soil microbes play an important role in plant litter degradation, thereby inducing changes in soil physical and chemical properties. On the other hand, soil microbes are affected by a suite of environmental factors, such as pH, temperature, and precipitation [4]. Thus, community diversities and functions as key indicators of soil microbial ecology have become hot topics in the field of ecological research in recent years [2,5,6,7,8]. Moreover, linking microbial communities to ecosystem functioning is a key challenge in microbial ecology [9]. The spatial distribution pattern of soil microbes, along with elevation, has been an important topic in microbial ecology research [10]. Plants and animals have obvious zonal and regional characteristics, as do soils, but it remains undetermined whether soil microorganisms have the same characteristics [4].

The ongoing rise in atmospheric CO_2_ concentrations represents a major environmental problem of recent years, driving increases in global temperatures [11]. Atmospheric CO_2_ is exchanged between the biosphere and the atmosphere, determining the direction and process of global climate change [12]; more than 80% of global carbon dioxide (CO_2_) emissions are due to the uncontrolled fermentation of organic matter [13]. Soil bacteria are key players in the global carbon cycle, capable of carbon fixation and carbon degradation [14]. It is necessary, therefore, to gain a deeper understanding of the structures and functions of soil bacterial communities in forest ecosystems to understand the response and adaptation of bacteria to global climate change and the feedback mechanisms in play.

The Qinling Mountains are a major east–west mountain range in the middle part of China. With high forest coverage, they play an important role in relieving global warming [15]. The interaction between species and environment in the Qinling Mountains forms unique habitats along an elevation gradient, which also leads to the formation of unique microbial diversity and structural features in soils [16]. Therefore, the Qinling Mountains are an ideal testbed for the comparative study of soil bacterial communities along elevation gradients. The aim of this study was to assess the diversity and structure of bacterial communities in three soil groups at different elevations using 16S rRNA gene amplicon sequencing. Microbial functional annotation of taxa was performed using the program “functional annotation of prokaryotic taxa” (FAPROTAX) based on OTU data, which may explain potential effects of bacterial communities on soil CO_2_ efflux. The results would allow for a better understanding of the patterns of soil bacterial communities and functional groups along altitudinal gradients and of the linkages between microbial communities and CO_2_ efflux. Our study revealed complicated patterns of soil bacterial responses to changes in conditions, highlighting the importance of taking bacterial functional traits into accounts when predicting soil carbon flux changes. By establishing linkages between functional gene abundances and soil CO_2_ fluxes, we provide novel insights into how bacterial functional traits could serve as potential proxies of ecosystem functions.

## 2. Materials and Methods

### 2.1. Site and Soil Sampling

The study area is located on the southern side of the Qinling Mountains, Shaanxi Province, China. A 100 m × 100 m site was set for each of the three sampling locations (Table 1, Figure 1), and 5 sampling plots were determined for each site for soil sample collection and CO_2_ efflux measurements. The sampling design is shown in Figure 1. From 1 to 3 July 2014, surface soil samples (0–10 cm) were taken in triplicate at each sampling plot using a soil borer with a diameter of 3.5 cm, and the triplicate soil samples were well mixed to give one sample with residual plants roots and debris removed. Fifteen soil samples were finally obtained, then placed in sterile plastic bags with aviation ice in a heat preservation box and transferred to the laboratory within 24 h. Samples were then stored at −80 ℃ in the laboratory for further analysis. Soil pHs were measured in situ using a portable pH detector (PH400, Spectrum, Aurora, CO, USA). Soil bulk density was measured using the core cutter method [17]. Elevations were measured using portal GPS (NAVA^®^ F30, BHCnav, Bejing, China). Annual temperature and annual precipitation were obtained from the WorldClim database [18]. The forest types of the A, B and C locations were *Quercus aliena*, *Larix kaempferi* and *Picea asperata*, respectively, with bamboo under the trees. The type of soils in the study area was brown forest soil—alfisol, according to Chinese soil taxonomy.

### 2.2. Soil DNA Extraction, PCR Amplification, and Illumina Sequencing

Soil DNA from samples was extracted using the CTAB/SDS method. The V4 hypervariable region of bacterial 16S rRNA genes was amplified by two rounds of PCR using a specific primer pair (515F-806R). Amplified products with bright main strips between 400–450 bp were chosen for further experiments. The chosen PCR products were mixed in equidensity ratios. Then, the mixed PCR products were purified with a Qiagen Gel Extraction Kit (Qiagen, Dusseldorf, Germany). Sequencing libraries were generated using TruSeq^®^ DNA PCR-Free Sample Preparation Kit (Illumina, San Diego, CA, USA) following the manufacture’s recommendations, and index codes were added. The library quality was assessed on the Qubit@ 2.0 Fluorometer (Thermo Scientific, Waltham, MA, USA) and the Agilent Bioanalyzer 2100 system (Agilent, Palo Alto, CA, USA). Finally, the library was sequenced on an IlluminaHiSeq2500 platform (Illumina, San Diego, CA, USA) and 250 bp paired-end reads were generated. Raw sequences were assigned to each sample using barcodes.

### 2.3. Raw Data Processing and Functional Annotation

Raw data processing was performed using Uparse software (Uparse v7.0.1001). Sequences with ≥97% similarity were assigned to the same OTUs. Representative bacterial OTU sequences were aligned for taxonomic annotation based on the RDP classifier (Version 2.2) against the Greengenes database. Sequences were rarefied to 70,000 reads per sample. Functional annotation of taxa was performed using the program “functional annotation pf prokaryotic taxa” (FAPROTAX) [19] on the rarefied OTU table. FAPROTAX is a manually constructed database that maps prokaryotic taxa to putative functions based on the literature on cultured representatives. It comes with a Python script for converting OTU tables into putative functional tables based on the taxa identified in a sample and their functional annotation in the FAPROTAX database.

### 2.4. CO_2_ Efflux Measurements

CO_2_ efflux was measured using static, manual polymethyl methacrylate chambers and gas chromatography techniques at the same time of soil sampling from 1–3 July, and it took one day to measure CO_2_ efflux at one sampling site. The chamber was an open-bottom square box (25 cm × 25 cm × 25 cm) with an electric fan installed on the top wall of each chamber to create turbulence when the chamber was closed. The chamber was covered with 2 cm-thick white foam to reduce the impact of direct radiative heating during sampling. A stainless steel base (25 cm × 25 cm × 20 cm) with a water-filled groove on top was installed at the three sampling sites. During the measurement, the chamber was placed over the base filled with water in the groove to ensure airtightness. Plants were removed before measurements were taken.

The measurement campaign involved setting up 9 chambers at three sites (3 chambers per plot; the measurement campaign lasted 3 days). Four air samples inside the chamber were collected every 15 min over a 45 min period using a 10 mL syringe equipped with three-way stopcocks every two hours from 09:00 to 17:00. Samples were injected into pre-evacuated packs, transported to the laboratory, and analyzed using gas chromatography (Agilent 7890A, Agilent, Palo Alto, CA, USA) equipped with FID and ECD within 36 h. The gas chromatography configurations for analyzing CO_2_ concentrations were determined according to previous studies [20], and the efflux calculation was performed following Song et al. [21].

### 2.5. Statistical Analyses

Coverage, Chao1 and Shannon’s index were calculated using QIIME (Version 2.0) and displayed with R software (Version 3.5.2) to reflect microbial diversity. The dissimilarity of within-group samples was calculated as weighted Bray–Curtis dissimilarity between pairs of replicates in a group. PCoA ordination based on Bray–Curtis dissimilarity was performed using the pcoa function in the Ape package. The Pearson correlation test was conducted with the rcorr function in the “Hmisc” package. Differences between groups were tested by one-way ANOVA. Partial Mantel tests were performed to detect the linkages between environmental variables and microbial taxonomic and functional communities. To determine the influences of microbes on carbon fluxes, multiple regression on weighted Bray–Curtis distance matrices was used to determine the influence of microbial functional composition and selected environmental variables on carbon effluxes, which was conducted with the trendline function in the “basicTrendline” package.

## 3. Results

### 3.1. Soil Bacterial Community Structure and α-Diversity

We obtained a total of 1,811,852 high-quality sequences and identified 38,606 OTUs (at the 3% evolutionary distance). The sequence number of each sampling site ranged from 74,815 to 238,702, from which 7290 to 9841 OTUs were recognized. The Good’s coverage of the OTUs in three sites ranged from 88.97% to 94.09% (Figure 2), indicating that the sequences sufficiently covered the diversity of bacterial populations in soil samples. 

Overall, microbial community α-diversity levels did not differ by sites (Figure 2). Chao1 of soil group C was the highest, followed by those of soil groups A and B. However, the greatest bacterial diversity measured by Shannon’s index was found in group A, with average values of 11.62, followed by group C, with average values of 11.44, and group B, with average values of 11.18 (Figure 2).

A total of 48 phyla were detected in this research. There were some differences among three soil groups in terms of dominant phyla (Figure 3). Sequences affiliated with *Proteobacteria*, *Acidobacteria*, *Actinobacteria*, and *Verrucomicrobia* were common in soil groups A and B, while the top four most abundant phyla in soil group C were *Proteobacteria*, *Acidobacteria*, *Verrucomicrobia*, and *Nitrospirae*. The phylum *Nitrospirae* is well-known to be closely related to soil nitrogen cycling.

### 3.2. Functional Group Abundances of Bacterial Communities

The top ten abundant functional groups of bacterial communities at the three sites are shown in Figure 4. Soil groups A and B had similar features, with chemoheterotrophic and aerobic chemoheterotrophic functions being the top two abundant functional groups, although their proportions in soil group A (56.82% on average) were higher than in soil group B (46.87% on average). However, functional group features of soil group C were significantly different from soil groups A and B (*p* < 0.01). In soil group C, nitrification (25.59% on average) and aerobic nitrite oxidation (19.38% on average) functional groups were higher than chemoheterotrophic (15.83% on average) and aerobic chemoheterotrophic (14.74% on average) functional groups (Appendix A).

We detected 12 functional groups associated with carbon degradation. The abundances of these functional groups decreased with increasing elevation, with average abundances being 61.66%, 51.91%, and 32.35% in soil A, B, and C, respectively (Appendix A).

### 3.3. CO_2_ Effluxes and the Associations with Bacterial Communities and Functional Composition

Similar to the distribution pattern of abundances of functional groups associated with carbon degradation, the results showed that soil CO_2_ effluxes decreased with increasing elevation (Figure 5). The CO_2_ effluxes of soil group A (0.30 g·m^−2^·h^−1^) were higher than those of soil groups B (0.27 g·m^−2^·h^−1^) and C (0.19 g·m^−2^·h^−1^) (Figure 5).

We tested whether there was a linkage between CO_2_ effluxes and soil bacterial community composition or functional groups composition. Principal coordinate analysis (PCoA) was conducted for bacterial communities and functional groups and correlated with the first axis of PCoA. It was found that the functional composition of bacterial communities (r = 0.59, *p* < 0.05; Figure 6a), rather than the taxonomic composition (r = 0.50, *p* = 0.058; Figure 6b), was significantly associated with soil CO_2_ effluxes. The PCoA results can be found in Appendix A.

Moreover, soil CO_2_ effluxes were significantly positively (r = 0.55, *p* < 0.05) correlated with the abundances of functional groups associated with carbon degradation (Figure 7). These results showed that bacterial functional composition was a better predictor of ecosystem functioning compared with bacterial taxonomic composition.

### 3.4. Effects of Environmental Factors on Soil Bacterial Communities and Their Functional Groups

We analyzed the effects of environmental factors on soil bacterial communities and their functional groups by means of partial Mantel tests. The effects of annual temperature and annual precipitation on bacterial taxonomic composition were significant (*p* < 0.01); bacterial functional composition (*p* < 0.05) and bacterial functional groups were associated with carbon degradation (*p* < 0.01) (Table 2). Moreover, the influences of pH on bacterial taxonomic composition (*p* < 0.05) and bacterial functional composition were significant (*p* < 0.05), but there was no significant correlation between pH and bacterial functional groups associated with carbon degradation (*p* = 0.509). There was no significant effect of soil bulk density on bacterial taxonomic composition (*p* > 0.05), bacterial functional composition (*p* > 0.05), or bacterial functional groups associated with carbon degradation (*p* > 0.05).

## 4. Discussion

### 4.1. Correlations between Environmental Factors, Bacterial α-Diversity and Taxonomic Composition

In our study, the high values for Shannon’s index (*H’* = 10.47–12.04) and richness (7290–9841) suggested the presence of a surprisingly high diversity of soil bacterial communities in the forest ecosystem of the Qinling Mountains. Du et al., reported a Shannon diversity of 8.66–8.69 for soil bacterial communities at a similar elevation on the south slope of the Qinling Mountains in summer [22]. By comparison, our results were more similar to the results of He et al., whose study was conducted on the north slope of the Qinling Mountains [23]. The results showed that soil bacteria could be easily influenced by microhabitats and a complex array of factors, such as plants, litter composition, and soil conditions; even animals would affect soil bacterial community [24]. What is more, the observed OTU richness and Shannon’s index in the Qinling Mountains in our study were much higher than those in regions such as Antarctica [8], which may be due to the higher temperature in our research area. Bacteria are temperature-sensitive; mid-latitudes are more suitable for bacterial survival [25]. It is well-recognized that microbial communities are spatially heterogeneous, and that the similarities between microbial communities decay with geographic distance.

Similar to the studies reported in [26,27], *Proteobacteria* was the most abundant bacterial phylum in soil environments recorded in our research. What is more, the relative abundances of the dominant phyla in our study were distinct among soils with diverse elevations. Notably, *Nitrospirae* was the fourth dominant phylum in soil group C, accounting for 8.27% of the population. However, it only ranked 3.22% and 3.28% in group A and B, respectively. These differences may be due to the factors previously found to be related to soil bacterial community, such as elevation [10,28], temperature [29], pH [30], moisture, and nutrient elements [31]. *Nitrospirae* in soil group C was highly related to the nitrogen cycle and elevation, which may reveal that the nitrogen cycle was more active at higher elevations [32]. What is more, the prevalent bacterial community would be changed due to soil conditions being transformed from aerobic to anaerobic states by inundation or other factors, anaerobic conditions boosting the growth of *Bacteroides* and *Euryarcheota* while discouraging *Firmicutes* and *Proteobacteria* [33].

Consistent with previous findings [1,34,35], we found that soil bacterial communities were significantly associated with soil pH. Bacterial α-diversity was higher in soils with acidic pH at the lower elevation in our study, similar to previous research [30,36].

Microbial taxonomic composition had no significant (*p* = 0.311) correlation with elevation in our study, differing from the results of Shen et al. [34]. Such a discrepancy could be attributed to the distribution of vegetation along elevation gradients, for vegetation type is a major factor in structuring soil microbial communities [37]. Vegetation types varied along elevation gradients in Shen’s research, while those in our study were the same. Moreover, the study which was also conducted in the Qinling Mountains similarly found no significant influence of elevation on soil bacterial communities [23].

Soil microbial communities and key soil processes are tightly linked to soil microclimate (temperature and moisture). Sierra et al., suggested that soil temperature was responsible for changes in bacterial community composition and activity [38], whereas other studies have shown that temperature-related environmental variables, such as soil water content, are the main factors in bacterial community dynamics [39]. In our research area with a mountain climate, annual temperature decreased along the elevation gradient while annual precipitation increased, which differed from the common pattern of synchronization of rain and heat. The interaction between temperature and precipitation may result in no significant correlation between elevation and soil bacterial communities [40].

### 4.2. Correlations between Environmental Factors and Bacterial Functional Composition of Soils

Linking microbial communities to ecosystem functioning is a key challenge in microbial ecology [9]. We used FAPROTAX to present functional phenotypes as metabolic and ecologically relevant functions in this research. Due to poor taxonomic identification of order and genus levels by 16s RNA sequencing, FAPROTAX cannot predict the function of all detected taxa [19], making it difficult to quantitatively discuss the deterministic relationship between soil bacterial functions and CO_2_ fluxes. However, the applicability of FAPROTAX has been verified by many studies [41,42,43]; FAPROTAX can be beneficial for FAST functional screening or grouping of 16S-derived bacterial data from forest ecosystems, and its performance has been enhanced through improving the taxonomic and functional reference databases [19].

We found that the functional composition of bacterial communities was not significantly associated with taxonomic composition; in fact, they were decoupled to some extent, which was similar to the results of Qi et al. [44]. Functional redundancy among microbial taxa, horizontal gene transfer, convergent evolution, or adaptive loss of microbial function could be responsible for the decoupling [45]. These results suggest that microbial functional traits are more sensitive indicators of environmental conditions than 16S rRNA gene-based microbial taxonomy, similar to the results of Ma et al. [46].

Moreover, environmental conditions can influence the distributions of functional groups in bacterial communities by shaping their metabolic niches in patterns that differ from those of bacterial taxonomic communities [7]. We found that bacterial functional structure was significantly correlated with environmental factors, including pH, elevation, annual temperature, and annual precipitation, similar to the results of Hirao et al [36,38].

In our study, the predominant functional groups of soil groups A and B were chemoheterotrophic and aerobic chemoheterotrophic, indicating that large numbers of bacteria obtained carbon and energy from the decomposition of soil organic matter [47]. Different environmental conditions could cause changes in bacterial community function in the soil [48]; the predominant functional groups of soil group C were nitrification and aerobic nitrite oxidation; the results may be caused by higher nitrogen contents in soil group C, for nitrogen deposition would occur more intensely in high-altitude areas [49].

The effect of pH on bacterial functional groups associated with carbon degradation was not significant, for temperature and soil moisture were the dominant factors affecting the enzyme activity of soil bacteria related to the degradation of available carbon rather than soil pH [50]. What is more, precipitation has a favorable/detrimental effect on microbes in changing soil water content, oxygen availability, and soluble substrate supply [51].

### 4.3. Effects of Soil Bacterial Communities and Bacterial Functional Groups on CO_2_ Effluxes

The amount of CO_2_ effluxes produced in different sites is an indicator of the activity of soil bacterial communities. Differences in bacterial diversity in soils would affect soil CO_2_ effluxes to some extent, but not as determinant factors with clear patterns [33]. On the contrary, the metabolic functions of soil bacteria determine soil CO_2_ emission fluxes [52], which is in accordance with our results. A strong positive and significant correlation between temperature or elevation and the metabolic functions of soil bacteria were found, from which it may be concluded that C turnover rate and CO_2_ flux would increase due to warming-driven selection [53].

In our study, the functional composition of bacterial communities, rather than the taxonomic composition, was significantly associated with soil CO_2_ effluxes, indicating that bacterial functions may be the key factors that explain ecosystem functioning. Moreover, bacterial functional composition was decoupled from bacterial taxonomy. Soil CO_2_ effluxes significantly increased with abundances of bacterial functional groups associated with carbon degradation, in contrast to the negative effect of whole functional groups of bacterial communities. The decomposition of specific carbon substrates could determine the different community compositions and functional groups among soil bacteria [54].

The abundance of bacterial functional groups associated with carbon degradation significantly decreased with elevation. The reason for this is twofold: firstly, the decreased average temperature at higher elevations may affect the numbers and activities of bacteria relevant to carbon degradation [55]; secondly, less soil organic matter and litter at higher elevations would also affect the abundances of bacterial functional groups [40].

What is more, both fungi and bacteria contribute to organic matter decomposition in soil [56]. In our research, only soil bacteria were analyzed and discussed for our research objective which was focused on soil bacterial communities along elevation gradients, however it may miss some key processes caused by soil fungi. This issue could be paid more attention through data analysis on fungal communities and the real functions of microbial communities, which could be obtained by ITS [57] and transcriptomics [58], respectively.

## 5. Conclusions

Our results showed that bacterial taxonomic composition had no significant correlation with elevation, though soil bacterial α-diversities changed with elevations. *Proteobacteria* and *Acidobacteria* were the dominant phyla in all soil samples, while *Nitrospirae* in soil group C was much higher than in soil groups A and B. Chemoheterotrophy and aerobic chemoheterotrophy were the two most abundant functional groups among the three soil groups. The functional groups identified in soil group C were also different from those identified in soil groups A and B. CO_2_ effluxes decreased with increasing elevation, while they increased with the abundances of bacterial functional groups associated with carbon degradation. Moreover, the functional composition of bacterial communities, rather than the taxonomic composition, was significantly associated with soil CO_2_ effluxes, suggesting that bacterial functional composition was decoupled from bacterial taxonomy. Annual temperature, annual precipitation, and pH had significant impacts on bacterial taxonomic communities, while elevation significantly affected bacterial functional groups. By establishing linkages between bacterial taxonomic communities, the abundances of bacterial functional groups, and soil CO_2_ effluxes, we have provided novel insights into how soil bacterial communities could serve as potential proxies of ecosystem functions.

## Figures and Tables

**Figure 1 microorganisms-10-00766-f001:**
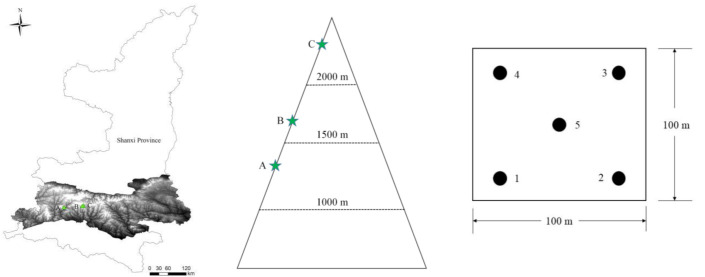
Locations of the three sampling sites and sampling design.

**Figure 2 microorganisms-10-00766-f002:**
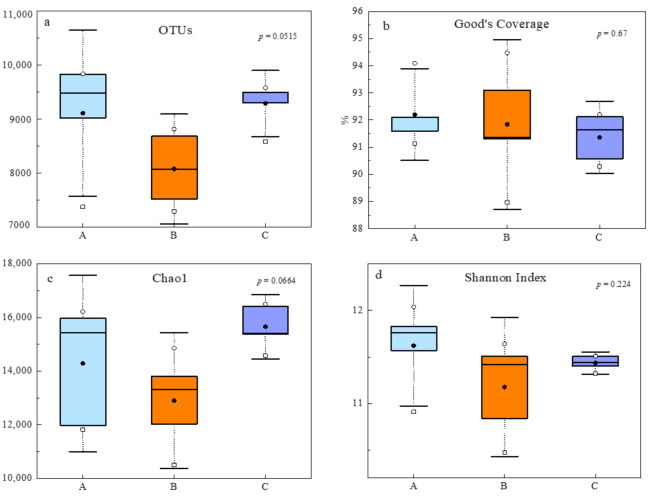
Basic information and α-diversities of microbial communities in soils. (**a**) OTU richness of different sites, defined at the cutoff of 3% sequence difference. (**b**) Good’s coverage, Coverage = 1− (number of individuals in species/total number of individuals). (**c**) Chao1, reflecting community species richness. (**d**) Shannon’s index, reflecting community diversity. Group statistical significance was assessed by one-way ANOVA followed by Tukey’s HSD test.

**Figure 3 microorganisms-10-00766-f003:**
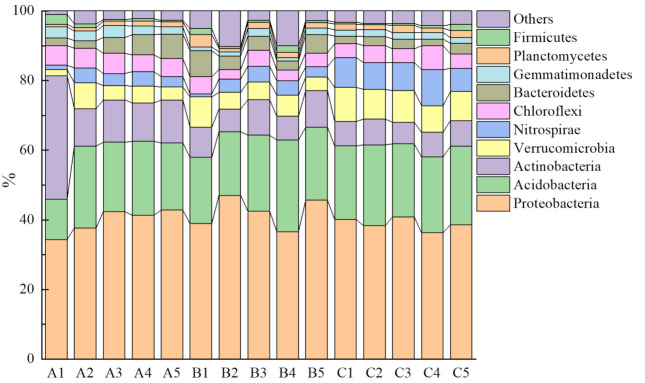
The relative abundances of the dominant bacterial phyla (top 10).

**Figure 4 microorganisms-10-00766-f004:**
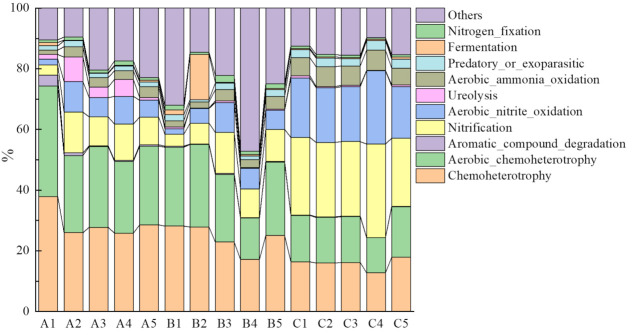
Abundance of functional groups of bacterial communities among soil samples.

**Figure 5 microorganisms-10-00766-f005:**
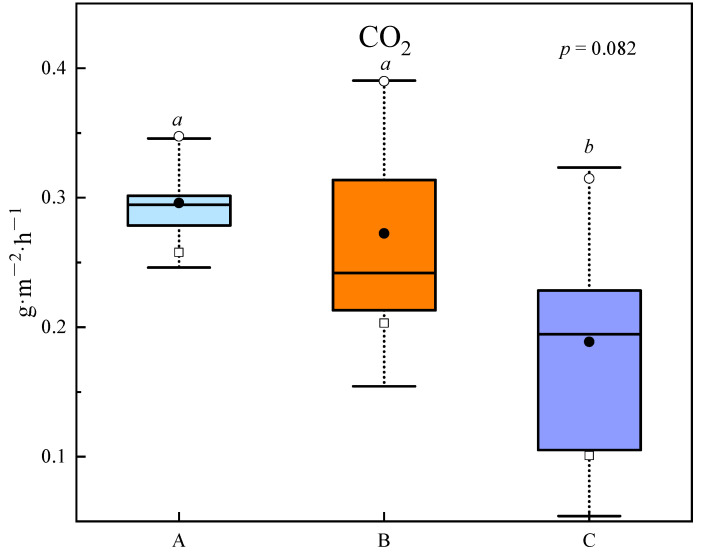
Daytime soil CO_2_ effluxes at different sites. (Different letters indicate significant differences among treatments (*p* < 0.05).)

**Figure 6 microorganisms-10-00766-f006:**
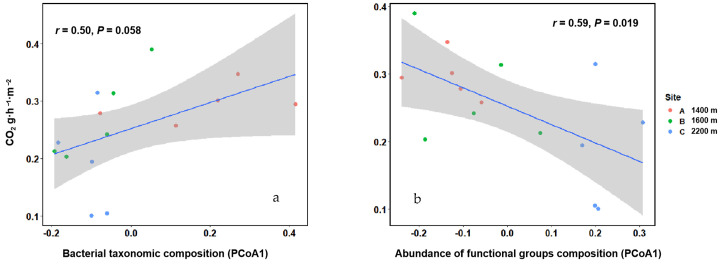
Correlations between CO_2_ effluxes and the bacterial composition and abundances of microbial functional groups with the first axis of PCoA. (**a**) Correlations between soil CO_2_ effluxes and bacterial taxonomic composition. (**b**) Correlations between soil CO_2_ effluxes and abundances of microbial functional groups composition.

**Figure 7 microorganisms-10-00766-f007:**
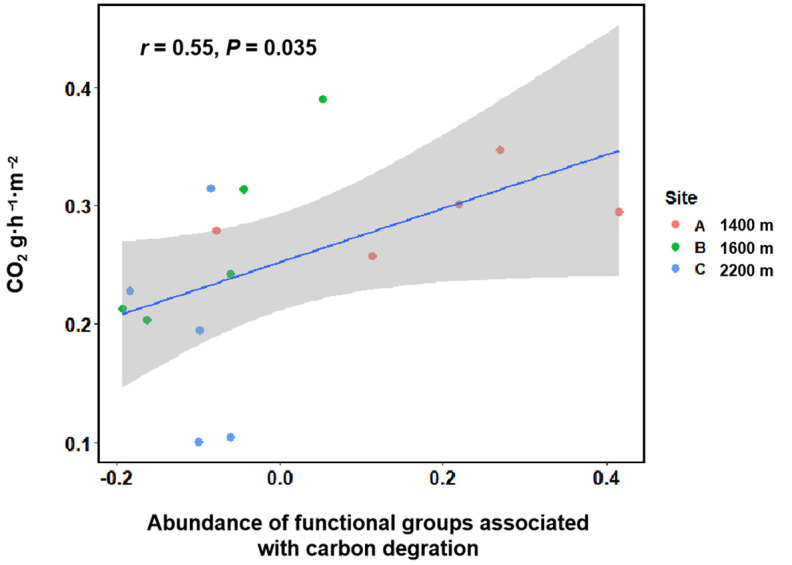
Correlation between soil CO_2_ effluxes and abundances of functional groups associated with carbon degradation.

**Table 1 microorganisms-10-00766-t001:** Location and geochemical properties of sampling sites.

Plots	pH	Elevation (m)	Annual Temperature (°C)	Annual Precipitation (mm)	Bulk Density (g·cm^−3^)
A1	4.10	1446.00	9.77	649.2	0.27
A2	3.99	1448.00	9.77	649.2	0.40
A3	4.19	1477.00	9.77	649.2	0.33
A4	3.99	1446.00	9.77	649.2	0.31
A5	4.18	1448.00	9.77	649.2	0.21
B1	4.55	1669.00	7.04	677.5	0.24
B2	4.55	1669.00	7.04	677.5	0.29
B3	4.52	1669.00	7.04	677.5	0.27
B4	4.66	1669.00	7.04	677.5	0.27
B5	4.50	1669.00	7.04	677.5	0.28
C1	4.05	2290.00	5.27	696.7	0.25
C2	4.16	2280.00	5.27	696.7	0.23
C3	4.11	2289.00	5.27	696.7	0.27
C4	4.00	2278.00	5.27	696.7	0.21
C5	4.13	2277.00	5.27	696.7	0.21

**Table 2 microorganisms-10-00766-t002:** The influence of environmental factors on bacterial communities and bacterial functional groups as determined by partial Mantel tests.

Variable ^a^	Bacterial Communities		Bacterial Functional Groups		Bacterial Functional Groups Associated with Carbon Degradation	
	*r*	*p*	*r*	*p*	*r*	*p*
pH	0.26	0.041 *	0.26	0.024 *	−0.02	0.509
Elevation	0.07	0.311	0.57	0.001 **	0.52	0.002 **
AT	0.32	0.009 **	0.26	0.021 *	0.38	0.005 **
AP	0.32	0.007 **	0.28	0.012 *	0.39	0.002 **
BD	0.05	0.315	−0.15	0.834	−0.07	0.621

^a^ Abbreviations: AT, annual temperature; AP, annual precipitation; BD, soil bulk density. *, *p* < 0.050; **, *p* < 0.010.

## Data Availability

Not applicable.

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
