# Peer review of "Bacterial Communities of Forest Soils along Different Elevations: Diversity, Structure, and Functional Composition with Potential Impacts on CO2 Emission"

_microorganisms, 2022, doi:10.3390/microorganisms10040766_

Round 1
Reviewer 1 Report
The manuscript by Sun et al. “Bacterial communities of forest soils along different elevations: diversity, structure, and functional composition with potential impacts on CO2 emission” requires revision to address major concerns.
Comments.
- Introduction (page 1) “In particular, bacteria ………. temperature, and precipitation [3]” The sentences can be restructured. As microbes like fungi in coordination with bacteria can play a significant role in plant growth i.e. doi : 10.3390/microorganisms9040774.
- Introduction (page 1) “research in recent years [2,4–10].” Many citations? Delete half of the citations.
- Page 2 “The ongoing rise in atmospheric CO2 concentration ………process of global climate change [14].” Please add information - i.e more than 80% carbon dioxide (CO2) emissions of the global emissions occurs by uncontrolled fermentation of organic matter (doi: 10.1016/j.rser.2021.111491). Both CO2 and methane are greenhouse gases (GHGs).
- Introduction, please state, how this study will be beneficial to counter CO2 rise in atmosphere and novelty of this study.
- Please avoid the use of standard deviation values in the text.
- Please add illustrations to understand the mechanism of microbial associations and the outcome of the present study data.
- Please discuss, the correlation between microbial diversity to forest soils and landfill-based system analysis for the GHGs emissions i.e. doi: 10.1007/s12088-021-00995-7.
- All Figures quality should be improved i.e. font sizes, line width, and resolution.
Author Response
Comments.
- Introduction (page 1) “In particular, bacteria ………. temperature, and precipitation [3]” The sentences can be restructured. As microbes like fungi in coordination with bacteria can play a significant role in plant growth i.e. doi : 10.3390/microorganisms9040774.
Answer: We restructured the sentence according the recommended reference.
- Introduction (page 1) “research in recent years [2,4–10].” Many citations? Delete half of the citations.
Answer:We have deleted three of the citations according to the reviewer’s comments. The original reference 5, 9 and 10 were deleted.
- Page 2 “The ongoing rise in atmospheric CO2concentration ………process of global climate change [14].” Please add information - i.e more than 80% carbon dioxide (CO2) emissions of the global emissions occurs by uncontrolled fermentation of organic matter (doi: 10.1016/j.rser.2021.111491). Both CO2 and methane are greenhouse gases (GHGs).
Answer: We have added the information which the reviewer recommended to the revised manuscript according the reference.
- Introduction, please state, how this study will be beneficial to counter CO2rise in atmosphere and novelty of this study.
Answer: We added the statements about the meaning of our study in the Introduction part “Our study revealed complicated patterns of soil bacterial responses to condition changes, highlighting the importance of taking bacterial functional traits into accounts when predicting soil carbon flux changes. By establishing linkages between functional gene abundances and soil CO2 fluxes, we provide novel insights into how bacterial functional traits could serve as potential proxies of ecosystem functions.”
- Please avoid the use of standard deviation values in the text.
Answer: We have revised the content where using standard deviation values in the text according the comments.
- Please add illustrations to understand the mechanism of microbial associations and the outcome of the present study data.
Answer: This is a rather constructive suggestion. Actually it’s better to construct some models such as SEM to show mechanism of microbial associations and the outcome, however poor taxonomic identification of order and genus levels was an important factor that limited FAPROTAX functional assignment in soil samples (Sansupa, et al,. 2011), resulting in difficult to quantitatively discuss the deterministic relationship between soil bacterial functions and CO2 fluxes. Even though, we added more illustrations in the discussion part to ensure the scientificity of research conclusions.
- Please discuss, the correlation between microbial diversity to forest soils and landfill-based system analysis for the GHGs emissions i.e. doi: 10.1007/s12088-021-00995-7.
Answer: We added discussions on the correlation between microbial diversity and forest soils according reviewer’s recommendation in the part “4.1. and 4.3”, which could descript as “What’s more, the prevalent bacterial community would be changed due to soil condition transformed from aerobic to anaerobic by inundation or other factors, while anaerobic condition boosted the growth of Bacteroides and Euryarcheota, whereas discouraged the Firmicutes and Proteobacteria” and “The amount of CO2 effluxes produced in different sites is an indicator of the activity of soil bacterial communities. Differences in bacterial diversity in soils would affect soil CO2 effluxes to some extent, but not the determinant factors with clear pattern. On the contrary, metabolic functions of soil bacterial were determine soil CO2 emission fluxes, which was similar to our result”.
- All Figures quality should be improved i.e. font sizes, line width, and resolution.
Answer: We improved the quality of figures in the revised manuscript according reviewer’s comments. The details could be found in the manuscript.
And what’s more, we improved the language of our manuscript by a native.

Reviewer 2 Report
Evaluation of the Ms 1648628 ”Bacterial communities of forest soils along different elevations: diversity, structure, and functional composition with potential impacts on CO2 emission” for Microorganisms by Sun et al.
The study compares the microbial community and functions associated to it along an elevation gradient. The study is nicely written in a logical way even though it needs a language check-up by a native.
Please rewrite the site and sampling chapter in your MM section. The reason is that it reads confusing since you use different words for the same thing. As an example, you say that you have three sampling sites along the elevation gradient and each site has five locations but in Table 1 you call the locations sites and reading further one does not know if you have taken 3 replicate soil samples per location (and bulked them) or per site (which would mean a statistical on of 3 per site and 9 in total). Call the 100x100 m area site and the 5 plots within each area plots. Then you have n=5 plots/site and then tell how you have sampled the plots and what is the final n for statistics.
Give a description of the soil type and vegetation for each site. Is the soil profile comparable at each site which allows you to take samples from all between 0-10 cm and compare them scientifically? This is actually a very relevant question if you can compare the results as you do.
In my opinion you should provide the soil C and N contents of the sites and include them in Table 1.
You report on the correlation of the microbial variables with the soil water content and the soil bulk density but soil water content measurement or the results are not presented (see Table 1). I also wonder why the bulk density is presented and "correlated" to the microbial community - what did you expect. Explain.
In your PCR approach-what means “Samples with bright main strip between 400-450 bp were chosen for further experiments”? And what does this mean for the final n?
CO2 – efflux measurement. Tell the date when you have performed the measurement and is it performed at the same time as the soil sampling for the microbial measures. As I understand you have measured the CO2 flux only once and you did not differentiate between heterotrophic and autotrophic respiration. This means that you did not measure microbial CO2 flux but still compare the result to the functional potential of the microbial community. I do not think this is ok and as an ecosystem measurement you must perform a longer measurement campaign than only taking one measurement.
Indicate the three elevation sites in Fig. 6. Is the Fig. 6 c functional composition also based on PCoA scores?
As you say correctly in the text FAPROTAX is a manually constructed database that maps prokaryotic taxa to putative functions based on the literature on cultured representatives. Your environmental samples compose of OTUs which to over 90% are not cultured and the real “species” or Genus are not known. Therefor the assumption of their function in the ecosystem is not based on facts. Still the results are presented and discussed to be the reality without any criticism. To get the real function of the community you should have used transcriptomics.
In the Introduction you argued for the need to gain better knowledge on the microbial influence on global warming in relation to the C turnover and CO2 flux. With the Ms at hand you do not answer this global question and you do not discuss it either. Also, the lack of the fungal community which strongly associates to the global CO2 cycle is missing. This could have been easily included in the sequencing work. I also miss a clear hypothesis.
Author Response
Comments and Suggestions for Authors
- The study compares the microbial community and functions associated to it along an elevation gradient. The study is nicely written in a logical way even though it needs a language check-up by a native.
Answer:We improved the language of our manuscript by a native.
- Please rewrite the site and sampling chapter in your MM section.The reason is that it reads confusing since you use different words for the same thing. As an example, you say that you have three sampling sites along the elevation gradient and each site has five locations but in Table 1 you call the locations sites and reading further one does not know if you have taken 3 replicate soil samples per location (and bulked them) or per site (which would mean a statistical on of 3 per site and 9 in total). Call the 100x100 m area site and the 5 plots within each area plots. Then you have n=5 plots/site and then tell how you have sampled the plots and what is the final n for statistics.
Answer:We rewrote the site and sampling chapter in the MM section. Where A、B and C as location, call the 100x100 m area site and the 5 plots within each area site. There are n=5 plots in one site, and finally there are 15 samples for statistics. And we also rewrote the detail information of soil sampling. Such as “surface soil samples (0-10cm) were taken in triplicate at each sampling plot using a soil borer with a diameter of 3.5 cm, and the triplicate soil samples were well mixed to one sample with residual plants roots and debris removed. 15 soil samples were obtained finally”.
- Give a description of the soil type and vegetation for each site.
Answer:We added the information of soil type and vegetation for A, B and C location, the details were “The forest types of A, B and C location were Quercus aliena, Larix kaempferi and Picea asperata, respectively, and with bamboo under the trees. The type of soils of study area was brown forest soil, alfisol according to Chinese soil taxonomy.”
- Is the soil profile comparable at each site which allows you to take samples from all between 0-10 cm and compare them scientifically? This is actually a very relevant question if you can compare the results as you do.
Answer:Comparability between samples was really important for obtaining scientific results. In our study, the soil profile was comparable at each site mainly for these four reasons: (1) the origin of the soil parent material in these sites is similar, and the soil types are the same as brown forest soil; (2) the thickness of the soil layers are relatively consistent, which were around 50-70cm, 0-10cm layer could represent top soil among these sites; (3) What’s more, the soil conditions of these sites were almost unaffected by human factors, the soil profile was not disturbed before be sampled; and (4) we carefully cleared up the litter and humus above the soil during sampling, and strictly followed the technical specifications for soil sampling to ensure the comparability between soil samples. Based on the above reasons, the soil profile is comparable at each site which allows us to take samples from all between 0-10 cm and compare them scientifically.
- In my opinion you should provide the soil C and N contents of the sites and include them in Table 1.
Answer:It’s a good suggestion that discussed more about correlation between soil nutrient elements and soil bacterial, or soil greenhouse gas fluxes. In terms of this research, we paid less attention to the effects of soil nutrient elements on the structure and function of soil bacterial communities, thus such physicochemical properties of soil were not analysed, which may confuse reviewers if there’re many factors without discussions. We will do further research focusing on relationships between nutrient element cycling and structure or function of soil microbial communities.
- You report on the correlation of the microbial variables with the soil water content and the soil bulk density but soil water content measurement or the results are not presented (see Table 1). I also wonder why the bulk density is presented and "correlated" to the microbial community - what did you expect. Explain.
Answer: We read our manuscript carefully with the question from reviewer, and found that it actually made somebody confusion without explanation on why soil water content was not presented. Firstly, soil water content and soil bulk density were important factors of soil condition which may influence microbial communities, thus we reported on the correlation of the microbial variables with them. But the data of soil water content were measured during soil sampling once, the data could not represent the true state of soil moisture content. For this reason, we did not take the measured soil water content to present or correlate, and take annual precipitation as instead which was more representative. In the revised manuscript, we revised the inaccurate description in the part “3.4” . What’s more, soil bulk density is relatively stable, thus we took it to present and correlate.
- In your PCR approach-what means “Samples with bright main strip between 400-450 bp were chosen for further experiments”? And what does this mean for the final?
Answer: The non-detailed description of the processing flow leads to the confusing sentences. As we knew that the prokaryotic 16S rRNA sequence contains 9 hypervariable regions, among which the V4 region has good specificity and complete database information, which is the best choice for bacterial diversity analysis and annotation. PCR amplification of V4 region could obtain 400-450 bp amplified products, the amplified products would be purified and generated sequencing libraries to be sequenced on an IlluminaHiSeq2500 platform. We rewrote this part according reviewer’s comments to make the description more accurate in part “2.2”. “Amplified products with bright main strip between 400-450 bp were chosen for further experiments. The chosen PCR products were mixed in equidensity ratios. Then, mixture PCR products was purified with Qiagen Gel Extraction Kit (Qiagen, Germany). Sequencing libraries were generated using TruSeq® DNA PCR-Free Smaple Preparation Kit (Illumina, USA) following manufacture’s recommendations and index codes were added. The library quality was assessed on the Qubit@ 2.0 Fluorometer (Thermo Scientific) and Agilent Bioanalyzer 2100 system. At last, the library was sequenced on an IlluminaHiSeq2500 platform and 250 bp paried-end reads were generated.”
- CO2efflux measurement. Tell the date when you have performed the measurement and is it performed at the same time as the soil sampling for the microbial measures. As I understand you have measured the CO2flux only once and you did not differentiate between heterotrophic and autotrophic respiration. This means that you did not measure microbial CO2 flux but still compare the result to the functional potential of the microbial community. I do not think this is ok and as an ecosystem measurement you must perform a longer measurement campaign than only taking one measurement.
Answer:
The sampling data had been told in Part “2.1”, and we added more information of sampling date in Part “2.1” and “2.4”, to describe CO2 efflux measurement performed at the same time as the soil sampling for the microbial measures.
Based on the theory of static box method, the measured CO2 flux was the total soil respiration flux per unit area at unit time, and actually could not differentiate heterotrophic and autotrophic respiration. However, plants were removed before measurements (as descript in the manuscript) to minimize the fluxes of autotrophic respiration, it made sense that comparing the result to the functional potential of the microbial community, similar researches have been done by other researchers such as Wei H (2018) in the Qinling Moutains.
And as the reviewer’s comments, a longer measurement campaign was needed to reveal spatial-temporal pattern between CO2 flux and the functional potential of the microbial community, and this is also we are doing. Nevertheless, one measurement under standard operation could also show meaningful results under the research objective, and providing phased conclusion for further research.
- Indicate the three elevation sites in Fig. 6. Is the Fig. 6 c functional composition also based on PCoA scores?
Answer: It’s a very good comment which could make the results more graphic. We added the three elevation sites in Fig. 6. And Fig. 6 c functional composition did not based on PCoA scores, In order to minimize the confusion, we divided Fig.6 to two Figures as Fig.6 and Fig.7 and rewrote the contents of this part.
- As you say correctly in the text FAPROTAX is a manually constructed database that maps prokaryotic taxa to putative functions based on the literature on cultured representatives. Your environmental samples compose of OTUs which to over 90% are not cultured and the real “species” or Genus are not known. Thereforethe assumption of their function in the ecosystem is not based on facts. Still the results are presented and discussed to be the reality without any criticism. To get the real function of the community you should have used transcriptomics.
Answer: This is a rather constructive suggestion. In truth, poor taxonomic identification of order and genus levels was an important factor that limited FAPROTAX functional assignment in soil samples (Sansupa, et al,. 2011)[3], thus we had to added the criticism of limitation of FAPROTAX used in the research before presented and discussed the results to ensure the scientificity of research conclusions. And as the comments of reviewer, transcriptomics should be used to get the more accurate and quantify function of the community.
According to the comments of reviewer, we rewrote the Part “4.2” to discuss the limitation of FAPROTAX first, then presented and discussed functional groups of soil bacteria. The results were also very meaningful even though the database cannot predict function of all detected taxa, it can be beneficial for fast-functional screening or grouping of 16S derived bacterial data from forest ecosystem. Thus the results from FAPROTAX were enough depend on its purposes.
The main revised part “We used FAPROTAX to present functional phenotypes as metabolic and ecologically relevant functions in this research. Due to poor taxonomic identification of order and genus levels by 16s RNA sequencing, FAPROTAX cannot predict function of all detected taxa [19], resulting in difficult to quantitatively discuss the deterministic relationship between soil bacterial functions and CO2 fluxes. However, the applicability of FAPROTAX has been verified by many studies [41-43], FAPROTAX can be beneficial for fast-functional screening or grouping of 16S derived bacterial data from forest ecosystem and its performance have been enhanced through improving the taxonomic and functional reference databases [19].”
- In the Introduction you argued for the need to gain better knowledge on the microbial influence on global warming in relation to the C turnover and CO2flux. With the Ms at hand you do not answer this global question and you do not discuss it either. Also, the lack of the fungal community which strongly associates to the global CO2cycle is missing. This could have been easily included in the sequencing work. I also miss a clear hypothesis.
Answer: We discussed the questions according reviewer’s comments in Part “4.3” to make the hypothesis of our research more clear.
The main revised part “The amount of CO2 effluxes produced in different sites is an indicator of the activity of soil bacterial communities. Differences in bacterial diversity in soils would affect soil CO2 effluxes to some extent, but not the determinant factors with clear pattern [33]. On the contrary, metabolic functions of soil bacterial were determine soil CO2 emission fluxes [52], which was similar to our result. A strong positive and significant correlation between temperature or elevation and metabolic functions of soil bacterial were found, which maybe concluded that C turnover rate and CO2 flux would increase due to warming-driven selection [53].” and “What’s more, both fungi and bacteria contribute to organic matter decomposition of soil []. In our research only soil bacteria were analysed and discussed which may miss some key processes caused by soil fungi [56], however the objective of this research was focused on soil bacterial communities along elevations, data of fungal communities and real function of microbial communities were paid no attention which could be obtained by ITS [57] and transcriptomics [58], respectively.”

Round 2
Reviewer 1 Report
Accept as is
Reviewer 2 Report
You have answered all my points. In your correction there were still linguistic faults and some still might occur in the text. Thus a language check by a native English speaking person is necessary